# *Meso*- and *Rac*-[bis(3-phenyl-6-*tert*-butylinden-1-yl)dimethylsilyl]zirconium Dichloride: Precatalysts for the Production of Differentiated Polyethylene Products with Enhanced Properties

**DOI:** 10.3390/polym14112217

**Published:** 2022-05-30

**Authors:** Kaitie A. Giffin, Virginie Cirriez, Orlando Santoro, Alexandre Welle, Evgueni Kirillov, Jean-François Carpentier

**Affiliations:** 1Centre National de la Recherche Scientifique (CNRS), Institut des Sciences Chimiques de Rennes (ISCR), University of Rennes, UMR 6226, F-35042 Rennes, France; kaitie.giffin@totalenergies.com (K.A.G.); orlando.santoro@uninsubria.it (O.S.); 2Total Energies One Tech Belgium, Zone Industrielle Feluy C, B-7181 Seneffe, Belgium; virginie.cirriez@totalenergies.com (V.C.); alexandre.welle@totalenergies.com (A.W.)

**Keywords:** *ansa*-zirconocene, *rac*/*meso* isomers, dual site, bimodal polyethylene, short-chain branches

## Abstract

*Ansa*-zirconocene complexes are widely employed as precatalysts for olefin polymerization. Their synthesis generally leads to mixtures of their *rac* and *meso* isomers, whose separation is often problematic. In this contribution, we report on the synthesis of a novel silyl-bridged bis(indenyl)-based metallocene, and on the separation of its *rac* and *meso* isomers by simple recrystallization from toluene. The two complexes, activated by methylaluminoxane (MAO), have been used as precatalysts in ethylene/1-hexene copolymerization. Regardless of the reaction conditions, the *meso* complex outperformed its *rac* congener. A similar trend was observed by performing the process in the presence of the silica-supported versions of the complexes. This is remarkable, since *meso* metallocenes generally display lower activities than their *rac* analogues. Furthermore, the *meso* isomer generates polymer products that are more in line with the targets for the preparation of a bimodal PE grade made of a lower-MW high-density (HDPE) fraction and a higher-MW linear low-density (LLDPE) fraction.

## 1. Introduction

In the field of commodity polyethylene (PE), there is an ongoing demand for polyethylene products with enhanced properties. Ziegler–Natta catalysis provides highly processable products (under certain conditions, such as injection molding) due to their broad molecular weight distribution (MWD) and normal comonomer incorporation (short-chain branches concentrated in the lower-molecular-weight chains). On the other hand, metallocene catalysts afford products with a narrow MWD and a uniform distribution of short-chain branches [1,2,3]. Such products can be advantageous on the basis of improved mechanical properties; however, they are often accompanied by processing issues. One solution to circumvent this shortage is the production of bimodal polyethylene with an inverse comonomer incorporation (short-chain branches concentrated in the higher-molecular-weight chains) [4,5]. The lower-molecular-weight (MW) component improves product processability, while the high-MW component is known to enhance mechanical properties (i.e., strength and stiffness). Bimodal PE grade made of a lower-MW high-density (HDPE) fraction and a higher-MW linear low-density (LLDPE) fraction is a highly desirable commodity polymer due to its improved performance relative to monomodal PE in a variety of applications, including its use in damage-resistant pipes and in lighter-weight flexible packaging [6].

For the copolymerization of ethylene with an α-olefin, further improvements to the state-of-the-art can be achieved with the design of new catalyst combinations that result in a higher-density split between the lower-MW and the higher-MW fractions (very low comonomer incorporation in the shorter chains vs. the longer chains). This can be achieved through the rational design of new metallocenes. In this context, there is great interest in discovering industrially relevant highly active metallocene catalysts that generate a low-molecular-weight PE product with very low comonomer incorporation.

When mixtures of *rac* and *meso* stereoisomers of a given *ansa*-bis(indenyl) zirconocene are used in the homo-/copolymerization of ethylene with an α-olefin comonomer (e.g., 1-hexene), a broader MWD and variable comonomer incorporation is observed in the generated polymer. From a practical point of view, these types of precatalyst mixtures can lead to a decrease in reproducibility and a lower degree of control in the design of the polymer architecture. The isolation of racemic metallocene from the *rac*/*meso* mixture is often targeted, as each stereoisomer generates homo-/copolymers with features distinct from one another, such as a specific MWD, comonomer incorporation, activity, and overall response to other reaction parameters. Previous investigations of *meso*- and *rac-ansa*-bis(indenyl) zirconocenes have typically found that the *rac* isomers are more active in ethylene polymerization and generate products with higher MWs compared to their *meso* counterparts [7,8]. 

In this study, we report the synthesis and isolation of both *rac* and *meso* complexes of a new silyl-bridged disubstituted bis(indenyl) zirconocene. Both isomers were tested in the copolymerization of ethylene/1-hexene, and the resulting copolymers were characterized on the basis of the MW, the MWD, and the comonomer incorporation. In contrast to previous studies [7,8], the *meso* isomer was found to be a significantly more active polymerization catalyst than the *rac* isomer. Furthermore, the *meso* isomer produces polymer products that are more in line with the targets outlined above for the low-MW high-density block of a bimodal copolymer. 

## 2. Materials and Methods

General considerations. Unless otherwise stated, all manipulations were conducted under nitrogen atmosphere, using either Schlenk or glovebox techniques. Solvents were dried on molecular sieves (4Å and 13X, 1:1 ratio) and activated alumina using a solvent purification system, deoxygenated by nitrogen purging and stored over activated 4 Å molecular sieves. Glassware was oven-dried at 150 °C for >2 h. The synthesis of proligand 1 was developed in our laboratories [9], while its upscaling was conducted at MCN Co. All other chemicals were obtained from Acros Organics (Geel, Belgium), Alfa Aesar (Ward Hill, MA, USA), and Sigma Aldrich (St. Louis, MO, USA) and used as received. Precatalysts *meso*-2 and *rac*-2 were supported onto silica (from PQ, D50: 40 µ) (0.4 wt% Zr) and MAO (30 wt% solution in toluene; contains ca. 10 wt% of free AlMe_3_), using previously reported procedure [9].

NMR spectra of all organic and organometallic compounds were recorded on a 400 MHz Bruker Avance instrument at room temperature in Teflon-valved NMR tubes. ^1^H and ^13^C NMR chemical shifts are reported in ppm vs. SiMe_4_ (0.00), as determined by reference to the residual solvent peak. ^13^C{^1^H} NMR spectroscopic analyses of PE samples were recorded on an AM-500 Bruker spectrometer equipped with a cryoprobe using the following conditions: solutions of ca. 200 mg of polymer in 1,2,4-trichlorobenzene/C_6_D_6_ (5:1 *v*/*v*) mixture at 135 °C in 10 mm tubes; inverse-gated experiment; pulse angle: 90°; delay = 30 s; acquisition time: 1.25 s; number of scans = 240. 

DSC measurements were performed on a SETARAM Instrumentation DSC 131 differential scan calorimeter at a heating rate of 10 °C/min; first and second runs were recorded after cooling to 30 °C; the melting and crystallization temperatures reported in tables were determined on the second run. 

GPC analyses of PE samples were carried out in 1,2,4-trichlorobenzene at 135 °C using polystyrene standards for universal calibration.

### 2.1. Synthesis of [Bis(3-phenyl-6-tert-butylinden-1-yl)dimethylsilyl]zirconium Dichloride (***2***)

In a 500 mL round-bottom flask, to a solution of dimethyl bis[(3-phenyl-6-tert-butylinden-1-yl)]silane (**1**, 9.7 g, 552.8 g/mol, 0.0176 mol) in toluene (130 mL) was added *n*-BuLi (22.0 mL of a 1.6 M solution in hexanes, 0.0351 mol) over the course of 15 min. The color first changed from clear orange to dark red, then to a cloudy brown-beige just after the end of the addition. The mixture was left to stir at room temperature for 24 h. In a second 500 mL round-bottom flask, ZrCl_4_ (4.1 g, 0.0176 mol) was suspended in toluene (50 mL). With stirring, THF (2.7 g, 0.0370 mol) was added dropwise over ca. 5 min. This reaction mixture was left to stir at room temperature for 2 h. The suspension of the dilithiated ligand was then added over the course of 15 min to the ZrCl_4_/THF mixture. Extra THF (ca. 2 mL) was used to wash the white solid off the walls of the ligand dianion flask and ensure complete transfer. Over the course of the addition, the color changed to cloudy dark orange. The resulting mixture was left to stir at room temperature for 18 h and then filtered over a 75 mL POR3 frit packed with Celite (dried in the oven for 3 days prior to use). The reaction flask and Celite was washed with extra toluene (ca. 40 mL, until no orange color remained on the Celite). The filtrate was concentrated under vacuum to ca. 200 mL; an orange precipitate started to form on the walls of the flask. The flask was well sealed using silicone grease and a glass stopper, shipped out of the glovebox, and stored at −35 °C for 20 h. At this point, a significant amount of orange solid had precipitated. The flask was then left at room temperature to defrost, prior to returning to the glovebox. The mixture was filtered over a 75 mL POR4 frit, collecting a bright-orange solid and a red-orange filtrate. The solid was washed with pentane (2 × 3 mL), then dried on the frit for ca. 1.5 h. The solid was then transferred to a vial for storage: *Fraction #1*, 2.58 g (21% yield). The filtrate was concentrated under vacuum in a 500 mL round-bottom flask until an orange precipitate began to form. The flask was sealed with a greased stopper, shipped out of the glovebox, and stored at −35 °C for 20 h. The flask was defrosted at room temperature, returned to the glovebox, and the mixture was filtered over a POR4 frit, collecting a second fraction of bright-orange solid and an orange filtrate. The solid was washed with pentane (2 × 3 mL) and was left to dry under vacuum on the frit for 2 h. The solid was then transferred to a vial for storage: *Fraction #2*, 446 mg (4% yield). The *meso* purity of each fraction was determined by ^1^H NMR spectroscopy. Fractions #1 and #2 had similar *meso* purities and could be combined, resulting in an overall yield of 25% with a 24:1 *meso*/*rac* ratio (see the Appendix A). 

Isomer *meso*-2: ^1^H NMR (400 MHz, CD_2_Cl_2_, 25 °C; Appendix A, top): δ 7.56 (dd, 2H, J = 9.2, Ar-*H*); 7.62 (s, 2H, Ar*-H*); 7.53-7.51 (m, 4H, Ar-*H*); 7.40 (m, 4H, Ar-*H*); 7.28 (m, 2H, Ar-*H*); 7.09 (dd, 2H, J = 9.2, Ar*-H*); 6.05 (s, 2H, Cp-*H*); 1.43 (s, 3H, Si-C*H*_3(*endo*)_); 1.25 (s, 18H, C(C*H*_3_)_3_); 0.94 (s, 3H, Si-C*H*_3(*exo*)_). ^13^C{^1^H} NMR (CD_2_Cl_2_, 100 MHz, 25 °C; Appendix A, bottom): δ 151.8, 138.0, 135.1, 133.6, 129.3, 128.3, 126.3, 124.7, 119.9, 117.9, 86.9, 35.0, 30.8, 21.1, −0.59, −4.3.

Evaporation of the solvent from the mother solution and further recrystallization from pentane at room temperature afforded a *rac*-enriched product (30% yield, 1:6 *meso*/*rac* ratio, as determined by ^1^H NMR spectroscopy, see the Appendix A). 

Isomer *rac*-**2**: ^1^H NMR (400 MHz, CD_2_Cl_2_, 25 °C; Appendix A, top): δ 7.69 (dd, 2H, J = 9.0 Hz, Ar-*H*); 7.60 (s, 2H, Ar*-H*); 7.50-7.49 (m, 4H, Ar-*H*); 7.38 (m, 4H, Ar-*H*); 7.30 (m, 2H, Ar-*H*); 7.10 (dd, 2H, J = 9.0, Ar*-H*); 6.19 (s, 2H, Cp-*H*); 1.28 (s, 18H, C(C*H*_3_)_3_); 1.17 (s, 6H, Si-C*H*_3_). ^13^C{^1^H} NMR (100 MHz, CD_2_Cl_2_, 25 °C; Appendix A, bottom): δ 152.3, 134.7, 131.8, 130.1, 129.2, 128.4, 127.3, 126.3, 125.9, 124.7, 118.6, 116.1, 87.1, 35.1, 30.9, −1.65.

APPI+-MS (toluene, *m*/*z*) [M]^+^ Calcd for [C_40_H_42_Cl_2_SiZr]^+^: 712.1473. Found: 712.1473 (see the Appendix A).

### 2.2. Ethylene (Co)Polymerization Reactions

#### 2.2.1. Homogeneous Conditions

Polymerization tests were performed in triplicates in a 24-slot high-throughput screening reactor in 50 mL glass vials. Under nitrogen atmosphere, each vial was equipped with a magnetic stir bar and loaded with *n*-heptane (25 mL), a toluene solution of the activated catalyst (200 μL, [Al_MAO_]/[Zr] = 1000, [Zr]_0_ = 10 µM), and the desired amount of 1-hexene. The vials were sealed with a crimp cap and introduced into the corresponding slots of the reactor, thermostated at 80 °C. The reactor was closed and pressurized with ethylene at the desired pressure (the gas was introduced in the vials through a needle piercing the septum of the crimp caps). The reaction was stopped after 15 min by venting the reactor. The polymer samples were collected and dried in air at room temperature for 16 h, and under reduced pressure at 50 °C for 3 h.

#### 2.2.2. Heterogeneous Conditions

The tests were performed in a parallel reactor system integrating six 130 mL stainless steel reactors equipped with a thermocouple, a pressure transducer, and constant-pressure regulator. Each reactor featured an antechamber. Each vessel was loaded with iso-butane (75 mL), the desired amount of 1-hexene (0–3.0 wt%), H_2_ (800 ppm), and ethylene (23.8 bar), and the temperature was equilibrated at 85 °C for 30 min. Each antechamber was charged with heptane (2 mL), the supported catalyst (2.0 mg), and the desired amount of TIBAL. The polymerizations were started by pressurizing these mixtures in the reactors, and were stopped after 1 h by venting the reactors. The polymer samples were collected and left to dry in air at room temperature overnight.

## 3. Results and Discussion

### 3.1. Metallocene Synthesis

The investigation was focused on silyl-bridged bis(indenyl)-based metallocenes, due to the industrial and academic relevance of such catalyst systems for olefin polymerization [10,11,12,13]. The new dimethylsilylene-bridged bis(indenyl) compound **1** was synthesized by using an adapted procedure based on a previously reported protocol [9]. The targeted zirconocene complexes ***meso*-2** and ***rac*-2** were synthesized by treating the dilithiated salts of proligand **1** with ZrCl_4_(THF)_2_ (formed in situ by the addition of two THF equivalents to ZrCl_4_ in toluene, Figure 1). This synthetic strategy is unselective towards either isomer and leads to a ca. 55/45 mixture of ***meso*/*rac*-2**, as determined by ^1^H NMR analysis (see the Appendix A). Due to the disparate solubility of the *rac* and *meso* isomers in toluene, it was possible to isolate the *meso* form by fractional crystallization in a 30% overall yield and with a *meso* purity of 96%. The work-up of the filtrate of this reaction affords a *rac*-enriched product (30% yield, 75% *rac* purity) [14]. 

Though all of our attempts to obtain crystals of ***meso*-2** and ***rac*-2** isomers suitable for X-ray analysis failed, their possible structures were modeled by DFT computations. These calculations suggested the ***meso*-2** isomer to be slightly more stable (by 0.8 kcal·mol^−^^1^). This energy difference with the ***rac*-2** form corresponds to a respective theoretical ratio of ca. 4:1 at room temperature. Note, however, that this minimal energy difference falls within the accuracy of DFT computations usually accepted (2–3 kcal·mol^−1^) and is therefore not inconsistent with the ca. 1:1 ratio observed experimentally. 

For a better representation and comparison of the overall structures and steric hindrance around the metal center in each isomer, a set of regular descriptors was used, including the percentage of buried %V_bur_ volume [15] and Ind_cent_–Zr–Ind_cent_ bite angles. The computed %V_free_ data (determined as %V_free_ = 100 − %V_bur_) provide a measurement of the space available in the first coordination sphere of the metal center; steric maps were generated for a selected series of complexes (Figure 2).

The geometrical descriptors %V_free_ and the Ind_cent_–Zr–Ind_cent_ angles in the molecules of both isomers are very similar. Therefore, essentially, the differences in the mutual organization of the corresponding quadrants in the metal coordination sphere may intrinsically influence the performances of these two isomers in ethylene/1-hexene copolymerization catalysis. 

### 3.2. Copolymerization of Ethylene and 1-Hexene Catalyzed by Rac- and Meso-2 in Homogeneous Conditions

The compounds ***meso***- and ***rac*-2**, activated by methylaluminoxane (MAO), were tested as catalysts in the ethylene/1-hexene copolymerization (Table 1). Regardless of the amount of comonomer employed, the *meso*-**2**/MAO system proved ca. 3 times more productive than the *rac*-**2**-based analogue. Upon increasing the amount of 1-hexene, a progressive improvement in the catalysts’ activity was observed. Such a beneficial comonomer effect is consistent with previous reports [16,17]. In the case of *meso*-**2**, higher comonomer concentrations led to a narrower MWD (from 5.7 to 3.1 for 0 wt% and 3.0 wt% of 1-hexene added, respectively), while, with *rac*-**2**, the MWD values were found in a very narrow range (3.3−3.5). The 1-hexene content in the final polymers was found to be proportional to the amount of comonomer employed in each run, with *rac*-**2** exhibiting a 1.5-fold higher comonomer incorporation ability than *meso*-**2**. Moreover, higher ethylene pressure proved to have a detrimental impact on 1-hexene incorporation only with the *rac*-based system (Table 1, cf. runs 9−10 and 11−12). 

^13^C NMR spectroscopy analysis of the polymers prepared with the *meso*-**2**/MAO system indicated the formation of ethyl branches, whose extent could be slightly reduced upon increasing the ethylene pressure (Table 1, cf. runs 1–2 and 9–10). The occurrence of this type of branching has been frequently observed with *meso*-type metallocenes (*vide infra*) and could originate from a regular chain-walking mechanism [18,19,20,21,22].

### 3.3. Copolymerization of Ethylene with 1-Hexene Catalyzed by Heterogeneous Silica-Supported Rac- and Meso-2

The silica-supported versions of both isomers of the metallocene, namely, **supp-*meso*-2** and **supp-*rac*-2**, were tested as catalysts in the ethylene/1-hexene copolymerization (Table 2). As observed under homogeneous conditions, **supp-*meso*-2** proved a much more productive polymerization catalyst than **supp-*rac*-2**, regardless of the amount of comonomer introduced. This is remarkable since, to date, only a limited number of *meso*-metallocenes have displayed higher catalytic activity than their corresponding *rac*-isomers [24,25]. It is tentatively proposed that such difference could arise either from a faster deactivation of the *rac* complex, possibly due to the formation of dormant heterobimetallic Al/Zr species [26,27,28], or to the occurrence, for the *meso* isomer, of a “stationary chain” polymerization mechanism, in which the monomer coordinates the active center always from the same (more accessible) side [24,29].

A screening with varying 1-hexene content was subsequently performed to further explore the comonomer incorporation ability of the two catalysts (entries 3−10). For **supp-*meso*-2**, no significant differences were observed in terms of the productivity and polymer properties (MW, MWD, and melting-temperature values) when the weight % of 1-hexene added was increased from 0 to 3.0 wt%. **Supp-*rac*-2** also proved capable of incorporating 1-hexene, albeit to a much lesser extent than its homogeneous version. The difference between the comonomer incorporation abilities of the homogeneous and supported versions of ***meso*-** and ***rac-*****2** could be explained in terms of the accessibility of the metal center. In fact, in the heterogenized system, one face of the metallocene is hindered by the support, which hampers the coordination/insertion of larger monomers (i.e., 1-hexene or macromonomers) [16,30]. 

The formation of ethyl branches was observed with both catalyst systems, albeit to a greater extent in the case of **supp*-meso*-2**. The origin of ethyl-branch formation and the metallocene/bis(indenyl) substituent effects on the ethyl-branch content has been studied in detail by Oliva et al.; their combined experimental/theoretical study supports a mechanism in which β-hydride transfer to the coordinated monomer, followed by insertion of the unsaturated chain into the generated Zr−C(ethyl) bond, is competitive with regular chain propagation [18,19,20,21,22].

## 4. Conclusions

The metalation of the silyl-bridged bis(indenyl) proligand **1** afforded an almost equimolar mixture of the corresponding dichlorozirconocene complexes, namely, the *rac*-**2** and *meso***-2** isomers. Although attempted selective synthetic approaches towards the *rac*- and *meso*-complexes were unsuccessful, the two isomers could be separated by recrystallization from toluene. When activated by methylaluminoxane, both complexes proved productive in the ethylene/1-hexene copolymerization, both under homogeneous and slurry conditions. Remarkably and unlike the common literature trends, *meso*-**2** was found to be ca. 3 times more productive than its racemic counterpart. This was tentatively accounted to a possible faster deactivation of *rac*-**2**, or to the occurrence of a “*stationary chain*” *polymerization mechanism* with *meso*-**2**. Further mechanistic studies along these lines are underway. Moreover, polymerization studies for dual-site catalyst combinations that incorporate the new *meso*-bis(indenyl) zirconocene are planned.

## Data Availability

Not applicable.

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
