# Peer review of "Meso- and Rac-[bis(3-phenyl-6-tert-butylinden-1-yl)dimethylsilyl]zirconium Dichloride: Precatalysts for the Production of Differentiated Polyethylene Products with Enhanced Properties"

_polymers, 2022, doi:10.3390/polym14112217_

Round 1

Reviewer 1 Report

This is a fine paper.  The area of ethylene polymerization is a crowded area and much of the research is not readily available given the tendency of many of the industrial groups to protect their intellectual property as trade secrets and the lower popularity of this area of research for academic groups studying polymer chemistry.   That said, this remains an interesting paper.  While it is unfortunate that crystal structures of the meso- and rac- silyl-bridged bis(indenyl zirconocene could not be obtained, the data provided are convincing evidence about the structures of the pre-catalysts.  It is not unexpected that the ethylene polymerization chemistry is different for the two diastereomeric catalysts.  However, the greater reactivity of the meso- catalyst is not consistent with the open literature I know.  This is noted by the authors.

The only minor issue I had was with the experimental.  In particular, the description of the precursor indenyl complexes improperly mixes up the terms solution and suspension.  That is a bit surprising given one of the major points of the paper is the difference in solubility of the rac- and meso- zirconocenes.  Specirfically, line 102 describes a cloudy grown-beige mixture.  If it is cloudy, it is a suspension, not a solution.  Thus, in line 106, they cannot be adding a solution of the dilithiated ligand to the ZrCl4 suspension unless there is a filtration step missing in the experimental.

I clearly support publication with this one minor revision.

Author Response

Re: As suggested, the phrase has been changed in the revised manuscript. The word “solution” in line 106 has been replaced with the more appropriate “suspension”.

Reviewer 2 Report

This manuscript deals with the synthesis of polyethylene and, respectively, related copolymers comprising a minor fraction of butyl groups due to copolymerization with 1-hexene. Some ethyl branches were also identified due to careful analysis of the obtained products. These products were obtained by catalytic conversion of the monomers ethylene and, respectively, 1-hexene, with two organometallic compounds acting as catalyst precursors. Remarkably, those catalyst precursors were stereoisomers which could be separated from a reaction mixture containing both compounds by simple crystallization. The two catalyst precursors were unequivocally identified by single crystal X-ray diffraction. Catalytic experiments were performed under homogeneous as well as under heterogeneous conditions with silica as catalyst precursor support. The experiments were performed with great care, and they are described very accurate and in great detail. The results are presented in a clearly arranged way in a table. The table also contains a lot of useful information on the characterization and properties of the products, such as the quantities of side groups, number and weight average molar mass and melting temperatures. The molar masses are moderate but this does not impair the value of the study as the results are clear and the discussion is sound. I do not have a major problem with this study and recommend publication without changes.

Author Response

Recommendation: Publish without changes.